# The formation and evolution of Titan's winter polar vortex

Nicholas A. Teanby [1], Bruno Bézard[2], Sandrine Vinatier[2], Melody Sylvestre[1], Conor A. Nixon[3], Patrick G.J. Irwin[4], Remco J. de Kok[5], Simon B. Calcutt[4] & F. Michael Flasar[3]

Saturn's largest moon Titan has a substantial nitrogen-methane atmosphere, with strong seasonal effects, including formation of winter polar vortices. Following Titan's 2009 northern spring equinox, peak solar heating moved to the northern hemisphere, initiating south-polar subsidence and winter polar vortex formation. Throughout 2010–2011, strengthening subsidence produced a mesospheric hot-spot and caused extreme enrichment of photochemically produced trace gases. However, in 2012 unexpected and rapid meso-spheric cooling was observed. Here we show extreme trace gas enrichment within the polar vortex dramatically increases mesospheric long-wave radiative cooling efficiency, causing unusually cold temperatures 2–6 years post-equinox. The long time-frame to reach a stable vortex configuration results from the high infrared opacity of Titan's trace gases and the relatively long atmospheric radiative time constant. Winter polar hot-spots have been observed on other planets, but detection of post-equinox cooling is so far unique to Titan.

[1] School of Earth Sciences, University of Bristol, Wills Memorial Building, Queens Road, Bristol BS8 1RJ, UK. [2] LESIA-Observatoire de Paris, PSL Research University, CNRS, Sorbonne Universités, UPMC Univ. Paris 06, Univ. Paris-Diderot, 92195 Meudon, France. [3] Planetary Systems Laboratory, NASA Goddard Space Flight Center, Greenbelt, MD 20771, USA. [4] Atmospheric, Oceanic and Planetary Physics, Department of Physics, University of Oxford, Clarendon Laboratory, Parks Road, Oxford OX1 3PU, UK. [5] Department of Physical Geography, Universiteit Utrecht, 3584 CS Utrecht, Netherlands. Correspondence and requests for materials should be addressed to N.A.T. (email: n.teanby@bristol.ac.uk)

The Cassini spacecraft orbited Saturn from 1 July 2004 to 15 September 2017 and in total made 127 Titan flybys, providing coverage of nearly half of Titan's 29.5-year orbit around the Sun. Saturn and Titan have an obliquity of 26.7°, which gives rise to pronounced seasonal effects. Cassini's observations cover early northern winter (2004) to northern summer solstice (2017) with northern spring equinox occurring on 11 August 2009. For the first time this unique time series allows us to observe the detailed formation and evolution of Titan's winter polar vortices.

Prior to Titan's 2009 northern spring equinox, Cassini observed a well developed northern winter polar vortex in the stratosphere and mesosphere, along with a meridional circulation dominated by a single pole-to-pole cell that ascended at southern and equatorial latitudes and subsided over the northern winter pole[1–3]. Strong vortex winds formed a mixing barrier and effectively isolated the polar air mass, permitting a distinct temperature and composition within the vortex[2,3]. Pre-equinox observations of the north polar region[2–12] show the mid-winter vortex is characterised by: trace gas enrichment due to subsidence from upper-atmosphere photochemical source regions[3,9–12], a cold lower stratosphere due to long-wave radiative cooling and a lack of insolation[2,3,13]; and a hot stratopause/mesosphere due to subsidence and adiabatic heating with a peak temperature of ~200 K[2,3,13]. These pre-equinox observations detail the nominal configuration of the mid-winter polar vortex, which is broadly consistent with results from numerical models[14–20].

Post-equinox, Titan's south pole began to enter winter, which provides an opportunity to observe vortex formation in the south. This complements previous observations of the well established mid-winter vortex in the north. Following equinox, the vertical circulation at the south pole reversed almost immediately[21,22]. The resulting south polar subsidence created a mesospheric hot-spot (~180 K at 300–400 km altitude) and high-altitude enrichment of trace gases. The reversal was accompanied by a two-cell transitional global circulation, with upwelling around the equator and subsidence at both poles, which persisted for two years before a fully reversed single circulation cell was established[22]. The

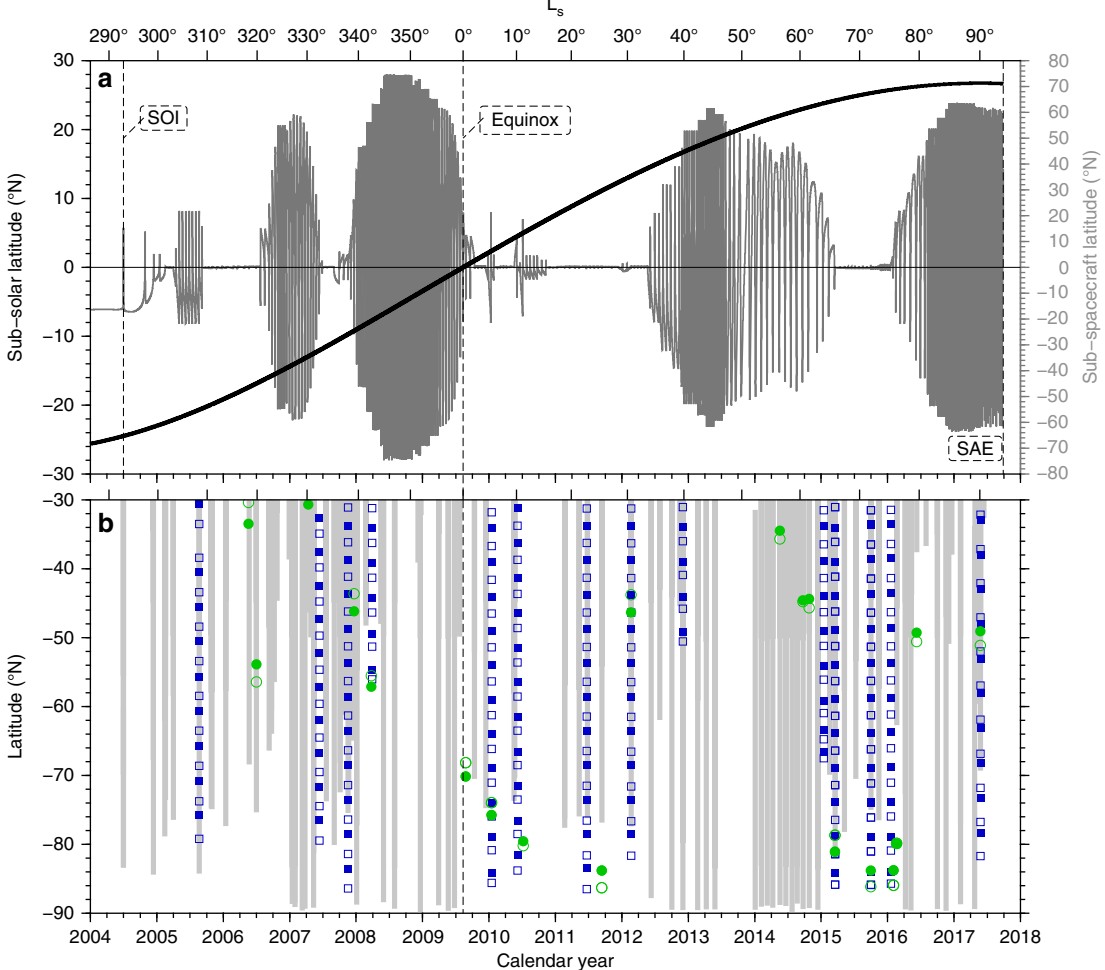

**Fig. 1** Cassini orbital information and CIRS data coverage. **a** Grey line shows the latitude of Cassini's sub-spacecraft point on Saturn as a function of calendar year and solar longitude ($L_s$). Cassini's complex orbit includes extended periods in Saturn's equatorial plane (sub-spacecraft latitude ~0°) to permit flybys of Titan and Saturn's other icy moons, but also includes significant portions with high orbital inclination to permit mapping of polar regions of Saturn, rings and moons. Thick black line shows evolution of sub-solar latitude during the Cassini mission, from southern summer at Saturn orbit insertion (SOI), through the northern spring equinox in August 2009, to northern summer solstice near saturn atmospheric entry (SAE) at the end of the mission in September 2017. **b** Coverage of southern hemisphere CIRS nadir and limb data. Nadir 2.5 cm$^{-1}$ mapping coverage shown by grey vertical bars; limb 0.5 cm$^{-1}$ integrations shown by solid/open green squares for FP3/FP4, respectively; and limb 14 cm$^{-1}$ cross-section maps shown by solid/open blue squares for FP3/FP4, respectively. Due to orbital constraints on viewing geometry, nadir and limb data have complementary coverage of the south polar region. The high spectral resolution 0.5 cm$^{-1}$ limb integrations (green) have the best signal-to-noise and altitude information, but have limited spatial and temporal coverage except around 80°S

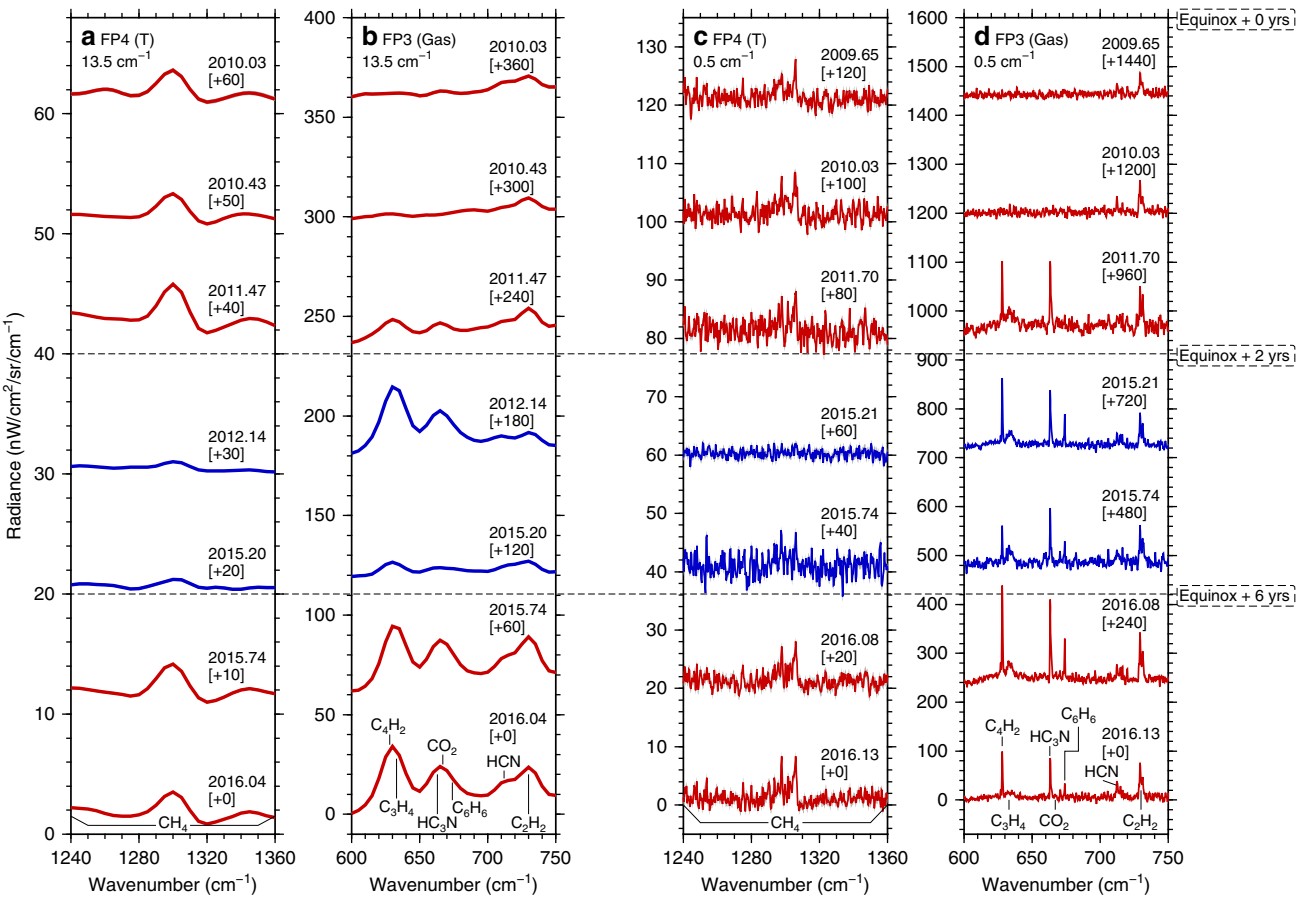

**Fig. 2** Post-equinox evolution of Cassini CIRS thermal emission spectra of Titan's south polar vortex from 2010 to 2016. CIRS spectra are taken in limb viewing mode and are measured at 80°S latitude and 400 km altitude (mesosphere). **a**, **b** 14 cm$^{-1}$ and **c**, **d** 0.5 cm$^{-1}$ spectral resolution. **a**, **c** The $\nu_4$ methane emission at 1300 cm$^{-1}$ is sensitive to temperature and indicates evolution from a hot mesosphere (2010–2011), to a cold mesosphere (2012–2015), and finally returning to a hot mesosphere in late 2015. **b**, **d** Emission from photochemically produced trace species HCN, HC$_3$N, C$_2$H$_2$, C$_3$H$_4$, C$_4$H$_2$ and C$_6$H$_6$ is significantly increased from 2011 onwards. Spectra are offset for clarity (offset given in square brackets). Note that the temporal coverage of CIRS observations is not continuous as it depends on Cassini's orbital geometry, so there is no limb coverage of the south pole in 2013 and 2014

timing of these circulation changes is consistent with numerical models[15–17,19,20], which subsequently predict a gradual evolution of the south polar vortex towards a state similar to that observed in the north prior to equinox.

However, contrary to expectations, in 2012 spectral features of HCN ice were identified at 300 km altitude over the southern winter pole with Cassini's Visible And Infrared Mapping Spectrometer (VIMS)[23]. This cloud was also observed in high spatial resolution images from Cassini's Imaging Science Subsystem (ISS)[24]. The presence of HCN ice is surprising as it requires temperatures of ~125 K to form at 300 km altitude—over 50 K colder than temperatures observed in mid-2011 at similar altitudes and latitudes[21–23]. Therefore, development of Titan's winter polar vortex is more complex than expected. There is not a simple monotonic increase of mesospheric temperatures following equinox to eventually mirror the ~200 K temperatures observed in northern winter at the start of the mission—instead rapid and unexpected cooling can occur.

Here we use 13 years of observations from Cassini's Composite Infrared Spectrometer (CIRS)[25] to explore the formation and evolution of Titan's south polar winter vortex. Primarily we use high vertical resolution limb (horizontal viewing) observations to determine the evolution of temperature and gas abundance during the post-equinox period. We supplement this with nadir (downward viewing) mapping observations to examine the horizontal extent of the vortex.

## Results

**Observation coverage**. Latitude-time coverage of the CIRS limb and nadir observations is shown in Fig. 1. Temporal coverage of the south pole with nadir data extend back to 2004 and allows a broad overview of south polar evolution, whereas high-resolution limb data of the south pole only exists from late 2009 onwards. The nadir and limb data coverage of the south pole is complementary, due to opposite polar viewing orbital requirements (Methods section).

**Limb results**. Initially, to investigate post-equinox mesospheric evolution we use CIRS limb spectra with tangent altitudes of ~400 km, which corresponds to the centre of the polar mesospheric hot-spot[21] (Fig. 2). Infrared emission from trace gases increases with both gas abundance and atmospheric temperature. Therefore, if we assume a uniform middle-atmosphere methane abundance, the $\nu_4$ band of methane centred around 1300 cm$^{-1}$ (7.7 μm) can be used as a thermometer, with high/low radiance corresponding to hot/cold temperatures, respectively. For the first 2 years after equinox, high mesospheric temperatures are indicated, as expected. However, in early 2012 the mesosphere exhibits a rapid cooling, evidenced by reduced CH$_4$ emission, which persists until 2015. Hot mesospheric temperatures return in late 2015, marking the end of a 4 year anomalously cold period. Since

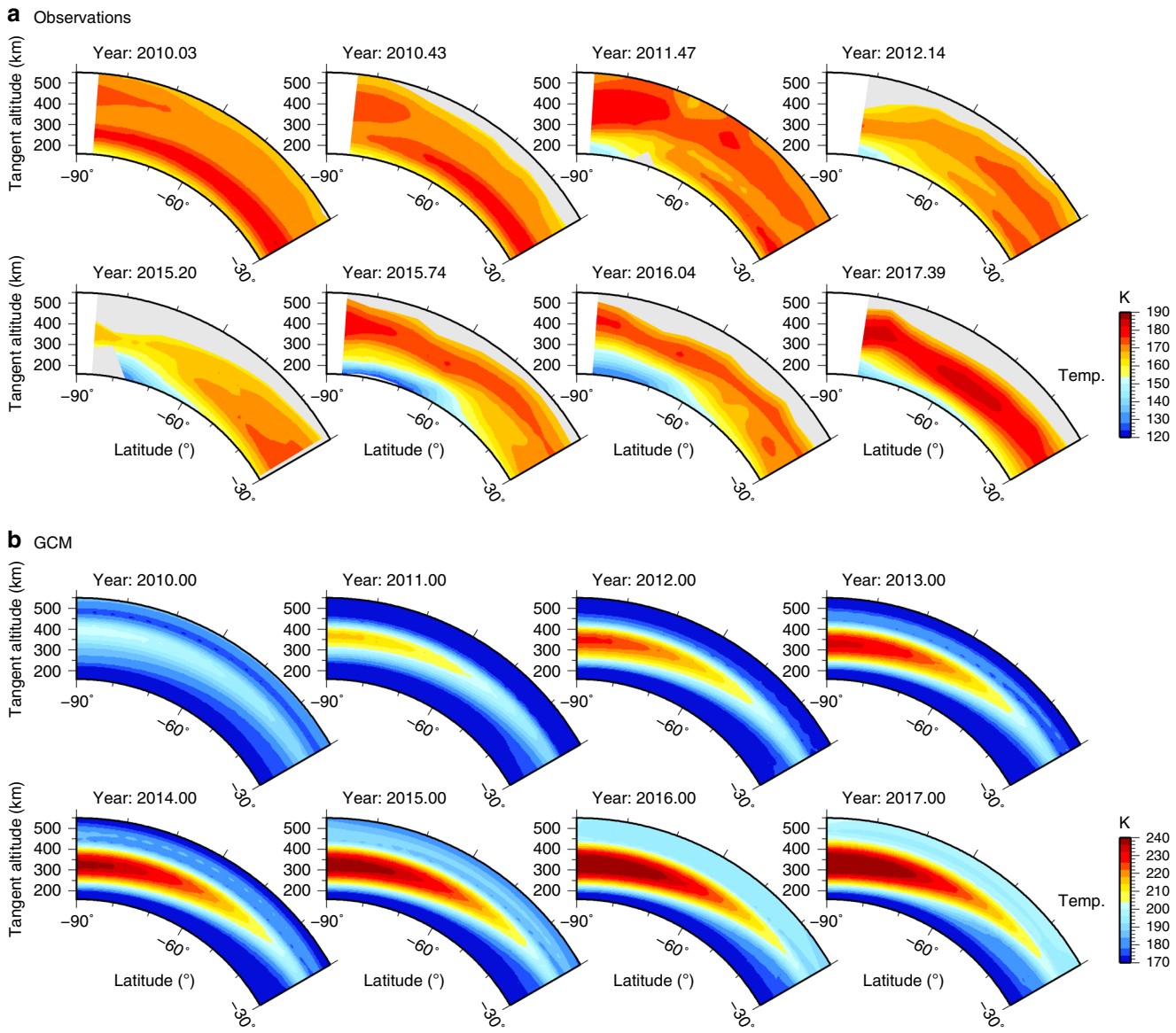

**Fig. 3** South-polar vortex temperature evolution. **a** Temperature cross-sections derived from Cassini CIRS 14 cm$^{-1}$ limb mapping observations. Grey areas indicate where signal-to-noise is too low to allow a reliable temperature determination. Observations clearly show development of an initial mesospheric hot-spot over the south pole at 400 km due to adiabatic heating, followed by a rapid cooling in 2012 which lasts for 4 years, and finally a returning hot-spot in late-2015. **b** Predictions from a general circulation model (GCM)[17] during the same period do not show the disappearance of the polar hot-spot. Note that the observations do not cover 2013 and 2014 due to orbital viewing constraints

2011, very high trace gas abundances are also inferred from these spectra, with enhanced emission from trace species HCN, HC$_3$N, C$_2$H$_2$, C$_3$H$_4$, C$_4$H$_2$ and C$_6$H$_6$ in the 600–750 cm$^{-1}$ wavenumber range, despite the significant temperature decrease. We also note that gases with short photochemical lifetimes[26–29] (HC$_3$N, C$_3$H$_4$, C$_4$H$_2$ and C$_6$H$_6$) show a more rapid increase in emission than gases with longer photochemical lifetimes (HCN, and C$_2$H$_2$). Gases with shorter photochemical lifetimes tend to have steeper vertical composition gradients, which results in greater subsidence-induced enrichment[8,9,30]. Therefore, even this simple inspection of the raw spectra shows that the south polar vortex takes at least 6 years after equinox to reach a stable temperature and composition.

In a more advanced analysis, we use radiative transfer combined with inverse theory[21,31] to quantify the evolution of temperature and composition in the south polar vortex from the observed spectra (Methods section).

First, we consider low spectral resolution (14 cm$^{-1}$) limb mapping observations. These have good latitude coverage of the polar region and can be used to obtain meridional temperature cross-sections, but cannot be used for detailed composition inversions as spectral lines of many important gases are blended together (Fig. 2). Figure 3 compares temperature cross-sections inverted from the limb mapping observations with predictions from a general circulation model (GCM)[17]. The observations clearly show the evolution of the vortex, which includes: progressive cooling of the lower stratosphere (100–250 km); initial growth of a mesospheric polar hot-spot (~400 km); sudden mesospheric cooling in 2012 by over 25 K; and reappearance of the mesospheric hot-spot in late-2015. No such mesospheric cooling is predicted by the GCM[17], which instead suggests continuous hot-spot development.

Second, we consider the high spectral resolution (0.5 cm$^{-1}$) limb integration observations. Each observation only covers a

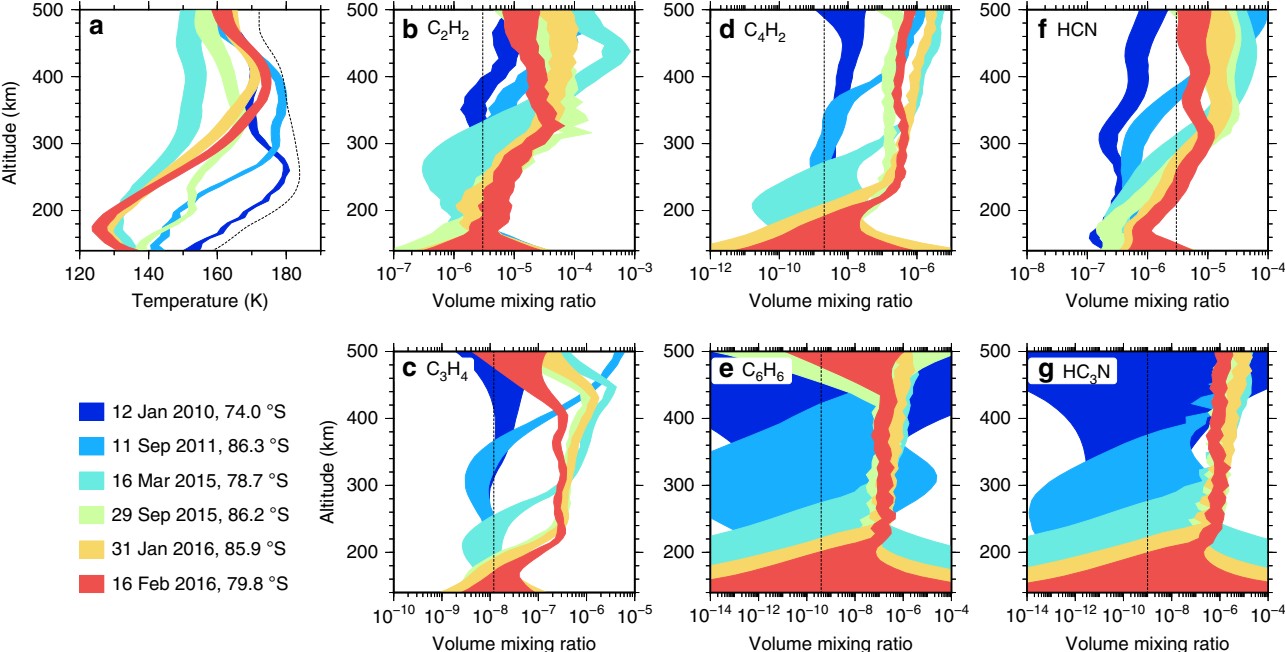

**Fig. 4** Evolution of south polar vortex temperature and trace gas abundance. Altitude profiles were determined from Cassini CIRS 0.5 cm$^{-1}$ spectral resolution limb integration observations at 80°S. **a** Middle-stratosphere and lower-stratosphere (100–250 km) temperatures generally exhibit a steady decrease until 2015, at which point the temperatures stabilise. Mesospheric temperatures at 400 km exhibit an early heating to 180 K, followed by a cooling of ≥25 K which persists until 2015, before heating up again to around 170 K in 2016. **b–g** Trace gas profiles are advected downwards during this period, leading to extreme enrichments of up to two orders of magnitude as air is brought down from the upper atmosphere photochemical source regions. Vortex composition stabilises around 2015, although there is a subsequent slight decrease in trace gas abundance at high altitude. Profile envelopes indicate 1-$\sigma$ inversion uncertainties. HC$_3$N and C$_6$H$_6$ have very large uncertainties for the earliest observations due to small initial abundances and resulting low signal-to-noise. Thin dotted lines show the temperature and trace gas a-priori profiles

single latitude, but with higher signal-to-noise than the mapping observations, and with a high enough spectral resolution to accurately determine abundances of individual gases. Figure 4 shows the mesosphere at 400 km initially heats up to 180 K in late 2011 at 80°S. The next observation in 2015 shows there has been a cooling of at least 25 K. Comparison with the limb mapping sequence in early 2012 (Fig. 3) shows this cooling must be rapid. By late 2015 the 400 km mesosphere has recovered and heated to 170 K, close to pre-cooling levels. In contrast, the middle-stratosphere and lower-stratosphere (~200 km) has gradually cooled from 170 K in 2010 to 130 K in 2015. The temperature of the lower stratosphere appears to stabilise in early 2015.

**Nadir results**. Nadir data are sensitive to lower atmospheric levels and can be used to place the higher altitude limb observations in context. Figure 5 shows example fits to the CIRS FP3 and FP4 nadir spectra at 80°S. Nadir spectra are most sensitive to composition and temperature in the 10–0.1 mbar pressure range (100–300 km), and provide evidence for cold stratospheric temperatures along with enhanced trace gas emission following the equinox. These data are less sensitive than limb data due to their shorter atmospheric path length, but have complementary temporal coverage, which shows that cold mesospheric temperatures persist during the gap in limb data coverage (2013–2014).

Figure 6 shows the evolution of stratospheric composition and temperature derived from the nadir data at the 1 mbar (~180 km) peak sensitivity level. The horizontal extent of the polar vortex can be seen to increase with time in both the stratospheric low temperature vortex core and polar trace gas enhancement. Gas enhancement is greatest in short lifetime gases, such as HC$_3$N and C$_4$H$_2$, which have become enriched by over two orders of magnitude since equinox.

**South-polar subsidence**. Subsidence development at the south pole is most clearly illustrated by the trace gas abundance profiles (Fig. 4), especially C$_3$H$_4$ and C$_4$H$_2$ where enrichment by downward advection is clearly visible. Assuming the gas profiles are purely controlled by advection of an initial photochemical profile, and ignoring other production/loss processes, gives estimated subsidence velocities of ~1 mm s$^{-1}$ between 2011 and 2015 for 300–400 km (0.01–0.05 mbar) and ~3 mm s$^{-1}$ between March and September 2015 for 200–260 km (0.1–0.5 mbar). Conservation of mass within the vortex core requires subsidence velocity to decrease with increasing pressure (decreasing altitude) for a given flow regime. Therefore, these measurements imply an increase in subsidence velocity throughout the stratosphere/mesosphere around 2015, coinciding with the mesospheric hot-spot return.

Since 2010 the enrichment of trace gases around the south pole has been extreme, with increases in mesospheric volume mixing ratios by up to two orders of magnitude. Abundances in the stratosphere stabilise around 2015 (Fig. 4). However, the 2015 and 2016 results suggest a slight abundance decrease at the highest altitudes for most species, which can be explained by an increase in vortex diameter with time causing the enriched gas to be diluted. Horizontal growth of the vortex is supported by nadir mapping data (Fig. 6), which shows a steady increase in vortex extent from 75–90°S in 2012 to 60–90°S in 2016 (taking the 140 K temperature contour in Fig. 6 as the vortex boundary).

## Discussion

We now consider the origin of the post-equinox mesospheric cooling. On Titan large abundances of radiatively active trace gases have the potential to significantly modify the energy balance within the south polar vortex. It is possible that enhanced radiative cooling from the extreme trace gas enrichments are

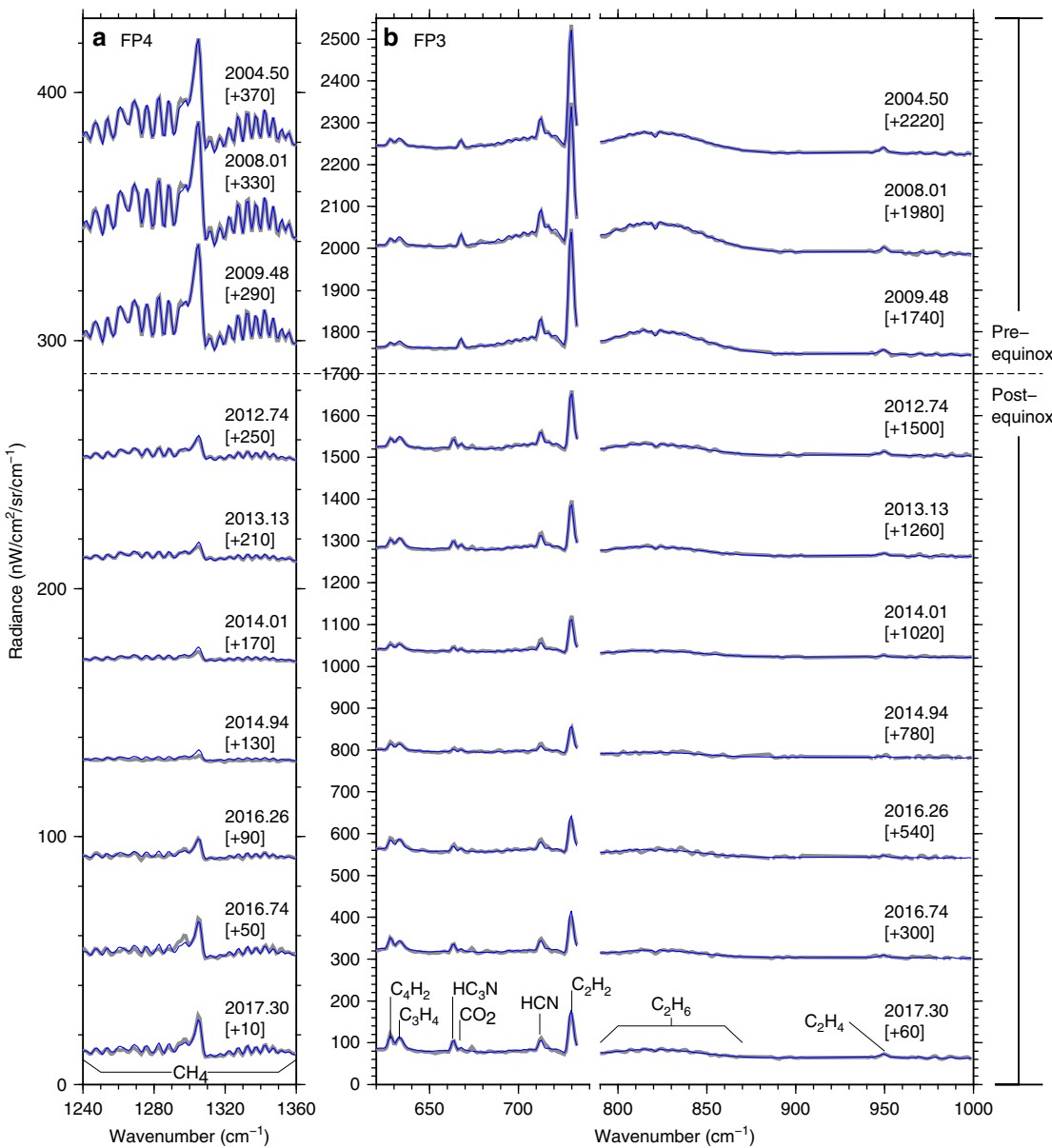

**Fig. 5** Evolution of Cassini CIRS nadir spectra of the south polar vortex. **a** FP4 spectra of the $\nu_4$ methane band, sensitive to stratospheric/mesospheric temperature. Pre equinox spectra show high emission and have a corresponding high temperature, whereas post-equinox spectra show minimal $\nu_4$ emission, indicating very cold stratospheric/mesospheric temperatures. From 2016 $\nu_4$ emission begins to increase indicating a return to a warm polar state. **b** FP3 spectra of trace gas emission peaks (labelled). Moderate trace gas emission enhancement is evident in the post-equinox spectra. For example, the HC$_3$N peak is more prominent from late 2012 onwards, despite the cold temperature prevalent at that time. All data were measured at 80°S. Blue lines show fits and grey lines with 1-$\sigma$ error envelopes show measured spectra. All data are fitted to within measurement uncertainty. Spectra are offset in radiance for clarity by the amount shown in square brackets

responsible for the cold mesospheric temperatures observed between 2012 and 2015. Even at mid-latitudes, where trace gas enrichment is much less extreme, long-wave radiative cooling has been shown to provide an explanation for temperature differences between 53°S and 50 °N observed by Voyager[32].

To quantitatively explore the effect of trace gases we develop a simple radiative balance model and consider the overall energy budget of the vortex. The rate of change of temperature $T$ at pressure level $p$ for a vertical subsidence velocity $w$ and southward meridional velocity $v$ is given by:

$$\frac{\partial T}{\partial t} = -r + w\left(\Gamma + \frac{\partial T}{\partial z}\right) + \frac{v}{(R+z)}\frac{\partial T}{\partial \theta} + s + \psi + \xi \quad (1)$$

where $r$ is the net long-wave radiative cooling rate, $w\Gamma$ is the adiabatic heating rate, $w\partial T/dz$ is the heating rate due to vertical advection of the ambient temperature profile, $(v/(R+z))\partial T/\partial \theta$ is the heating rate due to horizontal advection, $s$ is the solar heating rate, $\psi$ is the heating rate from atmospheric wave breaking, and $\xi$ is the heating rate due to horizontal mixing across the vortex boundary. Here $\Gamma = g/C_p$ is the adiabatic lapse rate for gravitational acceleration $g$ and atmospheric specific heat capacity $C_p$, $R$ is Titan's radius, $z$ is the altitude, and $\theta$ is the latitude.

The contribution of each of these heating/cooling terms depends on the altitude considered. For Titan's polar mesosphere and stratosphere, adiabatic heating due to subsidence is known to be significant[2] and provides a useful benchmark. The adiabatic lapse rate $\Gamma$ for Titan is $1.15 \times 10^{-3} \, \text{K m}^{-1}$, which for a typical

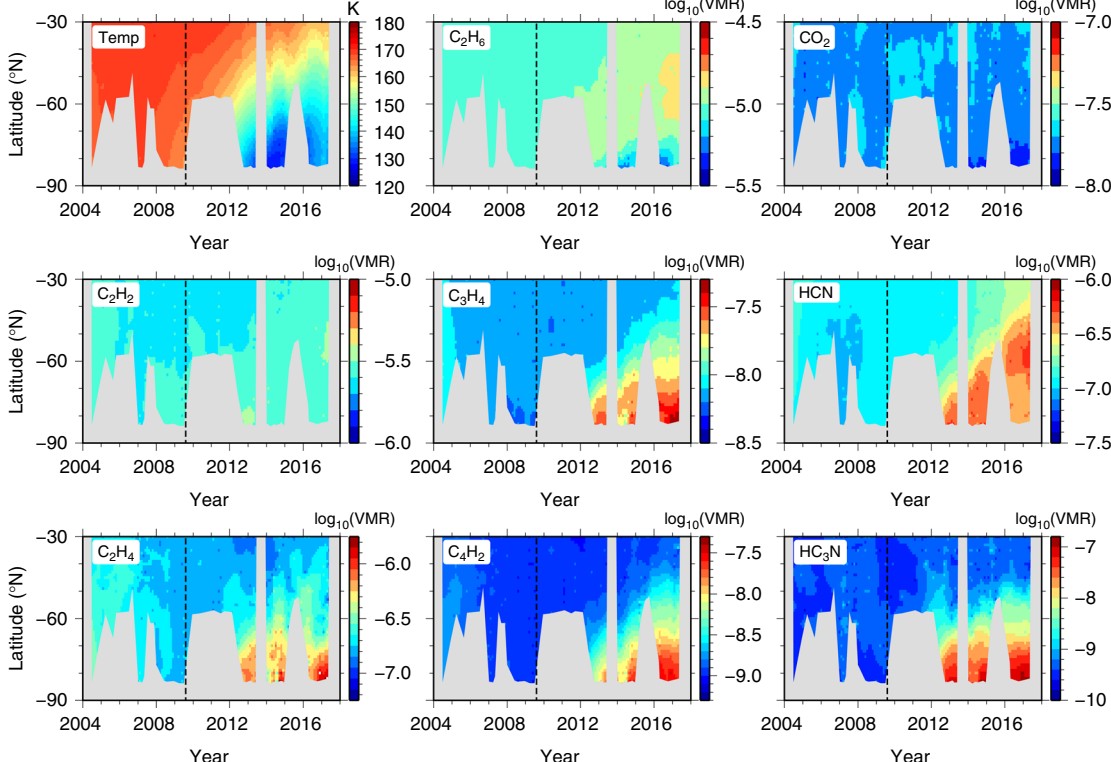

**Fig. 6** Temperature and composition evolution of the south polar region derived from nadir data. Results are shown for the 1 mbar pressure level (mid-stratosphere). The polar region exhibits significant cooling, which is accompanied by extreme trace gas enrichment, particularly for short lifetime species ($HC_3N$ and $C_4H_2$). The horizontal vortex size increases from ~75–90°S in 2012 to ~60–90°S in 2016. Grey areas indicate data gaps

mesospheric subsidence velocity of 0.005 ms$^{-1}$[21,22] corresponds to an adiabatic heating rate of 0.5 K per day. Therefore, to first order we assume that heating or cooling rate terms less than 0.1 K per day can be neglected.

Waves are expected to provide a significant heating source in the upper atmosphere >500 km (thermosphere and ionosphere)[33–35], which is consistent with wave-like temperature perturbations observed by the Huygens probe above 500 km[36] and Cassini's INMS instrument above ~1000 km[34,37]. However, in the stratosphere and mesosphere wave heating is expected to be small compared to other sources. For example[33], predict a nominal wave heating rate from tidal waves driven by Titan's orbital eccentricity to be ~$2 \times 10^{-10}$ Wm$^{-3}$ in the 250–500 km altitude range. This is equivalent to a heating rate of $\psi \leq 0.001$ K per day at 0.01 mbar (~400 km). However, there is also likely to be an unknown contribution from tropospherically generated gravity wave breaking or interaction of tropospherically forced planetary waves with the vortex. For example, on Earth sudden stratospheric polar warming can be generated by vertically propagating Rossby waves, which dissipate in the stratosphere and can change the strength and direction of the vortex winds, leading to sudden adjustments in the temperature field[38,39]. It is unclear if similar mechanisms are important on Titan and we do not consider them further here.

Heating by advection is also small for our altitude range of interest. The vertical temperature gradient measured in the 300–500 km range has a maximum value of $|\partial T/\partial z| \approx 0.2 \times 10^{-3}$ K m$^{-1}$ (Fig. 4). Therefore, the vertical advective heating rate will always be at least fives times less than the adiabatic heating rate and can be neglected. The horizontal temperature gradient in the south polar region can be measured from the limb (Fig. 3) and nadir (Fig. 6) temperature mapping. The maximum observed gradient is $(R + z)^{-1}\partial T/\partial \theta \approx 0.02 \times 10^{-3}$ K m$^{-1}$ in the mid-

stratosphere, although the gradient is much smaller near the stratopause. Using the general circulation model of refs.[15,40], the ratio of meridional to vertical velocities is expected to be $v/w \approx 5$ in the stratosphere and mesosphere in the winter polar vortex, indicating a somewhat convergent flow (a tentative poleward meridional flow was also derived from ISS cloud tracking observations at 300 km altitude[24]). Therefore, heating by horizontal advection ($5w\,\partial T/\partial x$) is at least 10 times smaller than that from vertical advection, so can also be neglected.

Finally, the polar vortex jet is predicted to be unstable on its equatorward side, which leads to the creation of planetary waves[40,41]. These waves can cause horizontal mixing, which acts to homogenise the temperature and trace gas distributions. The latitudinal transport is estimated to be greater than or equal to that due to direct meridional advection for stratospheric altitudes 150–400 km[15]. However, the process is less important above 400 km, where the meridional advection dominates[15]. Horizontal mixing will act to reduce the amplitude of any hot or cold mesospheric anomaly and so cannot be responsible for the anomalously cold region. Therefore, we neglect this term for simplicity.

By making these gross approximations and only including the dominant stratospheric and mesospheric heating sources, Eq. (1) simplifies to:

$$\frac{\partial T}{\partial t} \approx -r + w\Gamma + s \qquad (2)$$

To estimate the long-wave radiative cooling rate $r$ we use a plane-parallel atmospheric model following[42] (Methods section). Cooling rates calculated for atmospheric profiles measured in 2010, 2011, 2015 and 2016 are given in Table 1 and plotted in Fig. 7. We also calculated the cooling rates for the 2011 temperature profile with the composition from 2015 to determine the

**Table 1 Cooling rates for measured atmospheric profiles**

| $p$ | $z$ | $T(p)^a = 2010/01$<br>$C(p)^b = 2010/01$ | 2011/09<br>2011/09 | 2011/09<br>2015/03 | 2015/03<br>2015/03 | 2016/01<br>2016/01 |
|---|---|---|---|---|---|---|
| (mbar) | (km) | (K per day) | (K per day) | (K per day) | (K per day) | (K per day) |
| 0.01 | 382 | 0.75 | 1.6 | 3.3 | 1.2 | 2.4 |
| 0.10 | 268 | 0.50 | 0.50 | 0.55 | 0.11 | 0.34 |
| 1.00 | 175 | 0.17 | 0.06 | 0.07 | 0.04 | 0.06 |

Cooling rates calculated at the 0.01, 0.1 and 1 mbar pressure levels, where $p$ is the pressure and $z$ is the corresponding altitude. No high spectral resolution limb data were available for the south pole between 2011 and 2015, but using the 2015 composition profile with the 2011 temperature profile shows that extreme trace gas enrichment significantly increases the mesospheric cooling rate at 0.01 mbar
[a]Indicates year for the temperature profile $T(p)$
[b]Indicates year for the composition profile $C(p)$

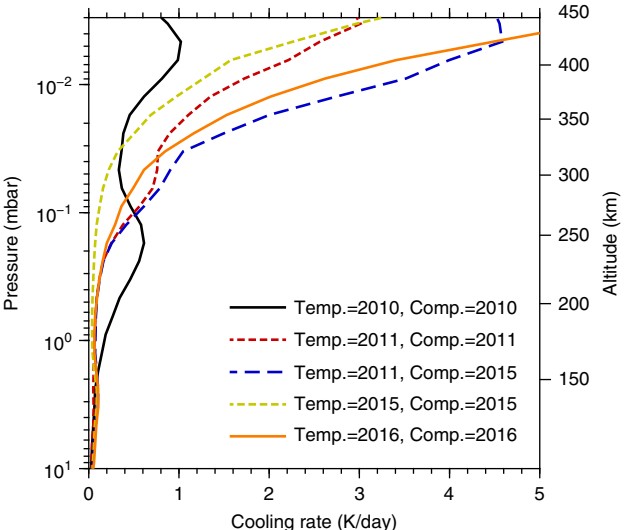

**Fig. 7** Long-wave radiative cooling rates calculated using the measured temperature and composition profiles within the south polar vortex. Calculations are for a latitude of 80°S. Mesospheric long-wave cooling is similar in 2011 and 2015. However, if the 2015 composition is used with the 2011 temperature profile the cooling rate is approximately a factor of two higher. This explains the cold mesospheric temperatures observed in 2012 in the low resolution limb data. The altitude scale is approximate and based on 2011 temperature profile

radiative effect of significant trace gas enrichment observed in 2015.

Our calculations show that the main contributors to long-wave cooling at the south pole depend on both the altitude and time after equinox. Just after equinox in 2010 and 2011 the main coolers are initially $C_2H_2$, $C_2H_6$ and $CH_4$ in the mesosphere at 0.01 mbar (~400 km). However, in 2015 and 2016 when the south polar mesosphere has become highly enriched in short lifetime trace gases, the main coolers change to $C_3H_4$, $C_4H_2$ and $HC_3N$, with minor contributions from $C_2H_2$, $C_2H_6$, $CH_4$ and HCN. At lower stratospheric altitudes (>0.1 mbar, <250 km) continuum cooling from aerosols becomes more important.

These calculations and approximate assumptions can now be used to draw broad conclusions about the evolution of the south polar stratosphere and mesosphere.

We first consider the mesosphere, which is typically hotter than the stratosphere (Fig. 4). Long-wave radiative cooling rates at 0.01 mbar (400 km) were calculated to be 0.75 K per day (2010), 1.6 K per day (2011), 1.2 K per day (2015), and 2.4 K per day (2016) (Table 1). Long-wave cooling is more efficient at high temperatures, so the atmospheric temperature will tend to

adjust towards an equilibrium state ($\partial T/\partial t \approx 0$) where the long-wave cooling is balanced by the combined adiabatic and solar heating. This quasi-balance is supported by the observation that overall temperature changes are <0.1 K per day at all altitudes (Fig. 4), which is small compared to typical diabatic and adiabatic heating rates (~1 K per day, Table 1).

In 2010, immediately after the August 2009 equinox, the meridional circulation had only just reversed and the circulation was relatively weak. Subsidence velocities at 400 km were estimated[21] to be ≈0.5 mm s$^{-1}$, which gives a low adiabatic heating rate of ≈0.05 K per day. This implies the long-wave cooling of 0.75 K per day must be almost entirely balanced by solar heating. We tested this assertion by estimating the solar heating rate for post-equinox orbital geometry. Our plane-parallel model is not ideally suited to calculating solar heating rates due to grazing incidence and lack of knowledge about visible haze opacity profiles in the polar region. Nevertheless, an approximate calculation based on nominal haze optical properties[43] implies a heating rate of ≈0.45 K per day at 0.01 mbar, which is consistent with the hypothesis that solar heating and long-wave radiative cooling are in approximate balance.

Conversely, in 2015 solar heating in the polar region can be assumed to be negligible due to shadowing from the surface and optically thick lower atmosphere. In this case the long-wave cooling of ≈1.2 K per day must be approximately balanced by adiabatic heating from subsidence. The long-wave cooling rates of 1.2, 0.11 and 0.04 K day$^{-1}$ at 0.01, 0.1 and 1 mbar (Table 1) then imply subsidence velocities of 12, 1.1 and 0.4 mm s$^{-1}$, respectively. This compares favourably to our estimate of ~3 mm s$^{-1}$ for 200–260 km (0.1–0.5 mbar) in 2015 from the composition profiles (Fig. 4).

Therefore, it is reasonable to assume that during 2010–2015 solar heating has a maximum value in 2010 and gradually decreases with time, whereas adiabatic heating is small in 2010 and gradually increases with time as the circulation develops. This suggests that adiabatic heating at 0.01 mbar cannot exceed 1.2 K per day and solar heating cannot exceed 0.75 K per day during 2010–2015, so the maximum possible heating rate is 1.9 K per day and is more likely to be ~1 K per day. However, if we combine the 2011 temperature profile with the 2015 chemical abundances, the calculated long-wave cooling rate dramatically increases to 3.3 K per day. This implies that the mesosphere must decrease its temperature to maintain the balance between long-wave cooling and the combined heating from insolation and adiabatic compression. Therefore, enhanced long-wave cooling from trace gas enrichment can explain the unusually cold mesospheric temperatures observed during 2012–2015. Our calculations show that at least half of the 25 K temperature decrease at 0.01 mbar is due to this composition change. In 2016 the cooling rate at 0.01 mbar is twice that in 2015 and the temperature has increased by ~20 K. Therefore, the subsidence

velocity must have also increased by a factor of ~2 between 2015 and 2016, also indicating the circulation is strengthening with time.

Stratospheric behaviour is conceptually simpler. From 2010 onwards, the lower- and mid-stratosphere continually cools, due to the reducing solar heating as it passes into Titan's shadow

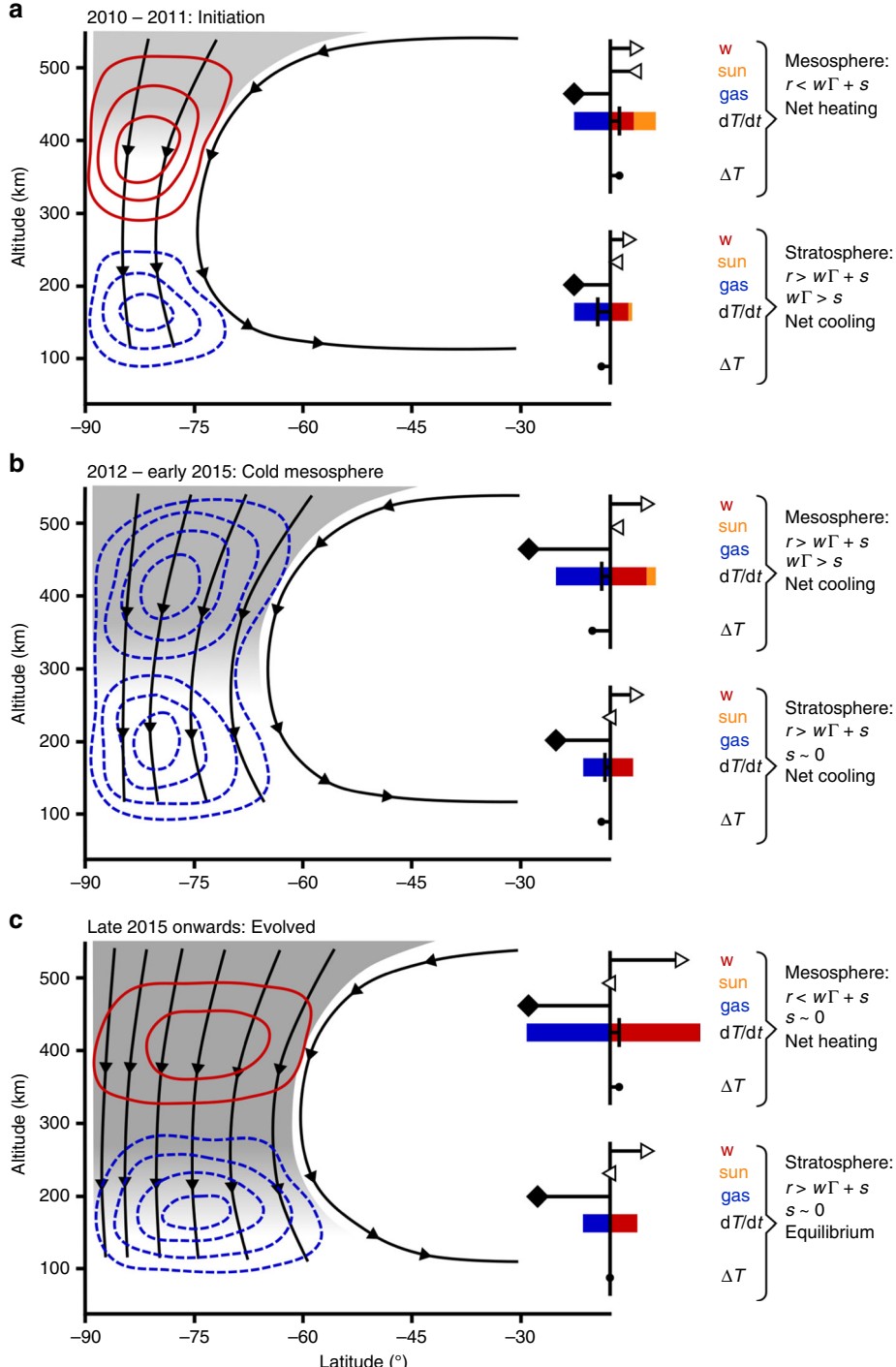

**Fig. 8** Schematic of the three stages of Titan's vortex evolution. Shaded areas represent trace gas abundance, which increases with time due to the subsiding flow (black streamlines). Red contours indicate hot anomalies and blue-dashed contours represent cold anomalies. Inset bar plots in each panel represent mesospheric and stratospheric heating and cooling processes; right-pointing triangles indicate subsidence speed $w$, which drives adiabatic heating rate $w\Gamma$ (red bar); left-pointing triangles indicate solar heating rate $s$ (orange bar); and diamonds indicate trace gas relative abundance, which drives long-wave radiative cooling rate $r$ (blue bar). The overall net heating rate $\partial T/\partial t = -r + w\Gamma + s$ (vertical line) produces either a hot or cold temperature anomaly $\Delta T$ (filled circle). Estimated long-wave cooling rates are given in Table 1 and range from 0.75 to 3.3 K per day in the mesosphere (~0.01 mbar, 400 km) and 0.04–0.17 K per day in the stratosphere (~1 mbar, 175 km). The three stages are: **a** vortex initiation, with a hot mesosphere driven by modest subsidence-induced adiabatic heating combined with weak cooling from trace gases; **b** cold mesosphere, caused by enhanced net cooling from increased trace gas opacity; and **c** evolved vortex, with a hot mesosphere driven by strong subsidence-induced adiabatic heating that exceeds enhanced cooling from trace gas enrichment

combined with enhanced long-wave cooling due to enrichment of photochemical gases and aerosols. Adiabatic heating in the stratosphere evidently does not increase enough with time to compensate for these two cooling effects, so the net effect is a gradual temperature decrease.

We conclude that vortex formation and evolution is controlled by three main processes: (1) long-wave radiative cooling, which increases in efficiency with both increasing trace gas abundance and increasing temperature; (2) insolation, which decreases with time as the south pole passes deeper into Titan's shadow; and (3) adiabatic heating, which increases with time as the circulation develops and subsidence velocity increases.

The high concentration of radiatively active gases in the polar vortex also raises the possibility of a dynamic-radiative feedback. Within the vortex, long-wave cooling is further enhanced due to dynamical isolation of the polar air mass, which prevents mixing with warmer sub-polar air and allows extreme enrichment of trace gases. For the middle- and lower-stratosphere (~100–250 km), the darkness of polar winter allows net long-wave cooling, which creates dense cold air and is the main driver of winter polar subsidence. It is also possible that the enrichment of radiatively active photochemical species enhances the thermal forcing of the circulation pattern by further decreasing the equilibrium temperature, which could strengthen the amplitude of the meridional circulation and in particular the polar subsidence.

The thermal and compositional structure of the winter polar stratosphere and mesosphere results from a complex balance between decreasing insolation, increasing subsidence, and increasing concentration of infrared radiators. We conclude that winter polar vortex formation on Titan can be characterised by three distinct evolutionary stages (shown schematically in Fig. 8).

The first stage is vortex initiation (2010–2011). Soon after equinox, reduced insolation and long-wave radiative cooling in the stratosphere create a latitudinal temperature and pressure gradient, which draws air towards the pole that converges and subsides to balance the cooling. The poleward moving air is spun up by the Coriolis effect, which results in the formation of a polar vortex. This isolates the polar air mass and causes further stratospheric cooling and subsidence. To conserve mass, subsidence velocity increases with altitude, so in the mesosphere subsidence is strong enough to cause sufficient adiabatic heating to overcome long-wave cooling, resulting in a mesospheric hot-spot around 400 km altitude (0.01 mbar). This first appears as a small perturbation in the 2010 temperature profiles and becomes obvious by 2011. Enhancement of trace gases due to downward advection is evident at high altitude[21,22].

The second stage has a cold mesosphere (2012–2015). Long-wave radiative cooling of the lower stratosphere continues, which further increases the subsidence velocity. When the reversed circulation has persisted for ~2 years, the mesosphere becomes highly enriched in short lifetime trace gases due to advection from upper atmosphere source regions. On Titan these gases are radiatively active and strongly increase the cooling efficiency of the atmosphere. This leads to a rapid (<6 months) decrease in mesospheric temperature to maintain the quasi-balance between heating and cooling rates. Cold mesosphere temperatures explain observations of the HCN ice cloud in 2012[23,24]. They could also contribute to strengthening the subsidence via a dynamic-radiative feedback.

The third stage is an evolved vortex (2015 onwards). Vortex composition stabilises, with horizontal vortex growth leading to a slight reduction in trace gas relative abundance. Subsidence velocity is now high enough that adiabatic heating outstrips the enhanced cooling efficiency due to trace gas enrichment, resulting in the re-appearance of a mesospheric hot-spot.

To conclude, our observations show that vortex formation on Titan is more complicated than previously thought, with trace gas enrichment playing an important role. We show there is an intimate link between dynamics and chemistry—not only can dynamics affect composition by advection of photochemical profiles[3], but composition can also affect dynamics through enhanced long-wave radiative cooling and the effects can be very long-lived. Post-equinox cooling perturbs the temperature structure from the expected state for 4 years, meaning that full vortex evolution takes much longer than predicted by numerical models[15–17,19,20]. The 4-year period of extremely cold upper mesospheric temperatures is not reproduced by current models and is not observed on other solar system bodies[44–46].

Enhanced thermal emission from trace gas enrichment has a major effect on the polar energy budget, especially in the mesosphere, and it is essential that future general circulation models include these effects. For computational efficiency GCMs only include the most important processes[47]. Full 3D models are computationally expensive and typically assume a uniform composition with latitude[14,19], although the most recent LMD model includes a zonally averaged latitude dependent photochemistry[20]. Axisymmetric 2D models allow an increased number of processes to be included[15–17,48]. These simulations have shown the radiative effect of haze is important, but polar enhancement of gases was considered a second order effect[15,16,48]. While this is valid for the stratosphere, our observations show that this is not the case for the mesosphere where a radiative-dynamic trace gas feedback is important. The difference stems from the fact that in GCMs where the radiative effect of gas enhancement at the pole has been taken into account the enhancements are much too small compared to what we observe. For example, the maximum polar gas enrichment in the model of[15] was ~10, based on early Voyager observations of the stratosphere[49]. Our new observations show that mesospheric enhancement at the pole can be over two orders of magnitude for some species, producing a much larger effect, especially in short lifetime species. Methane combined with long lifetime species with limited or modest enhancements ($C_2H_2$, $C_2H_6$) dominate the radiative cooling early in the vortex forming process, but are replaced by extremely enriched short lifetime species ($C_3H_4$, $C_2H_4$ and $HC_3N$) later in the season. Extreme enrichment of these short lifetime species is the main effect missing from current GCMs. Efficient mesospheric cooling coupled with relatively weak subsidence causes the mesospheric cooling. Only later in the season is the circulation strong enough for adiabatic heating to outstrip the enhanced cooling effect. It would also be beneficial for future GCMs to increase the model top altitude to above the currently used 500 km, so that more of the mesospheric chemistry and dynamics can be captured.

Titan is unique in the solar system because the major atmospheric coolers are photochemically produced in the high atmosphere, which means that subsidence within the winter polar vortex can significantly cool the mesosphere. This does not happen on the other terrestrial planets—Earth, Venus, and Mars—as the major atmospheric cooler on those planets is $CO_2$, which is uniformly mixed so is not enhanced by subsidence at the winter pole. The relatively low temperature of Titan's polar atmosphere (compared to Earth's) leads to a relatively large radiative time constant $\tau_{rad}$. This results in a slow advective timescale $(H/w \sim \tau_{dyn})$, which means stabilising vortex composition after initiation takes a long time as photochemical species must be advected from the upper-atmosphere source region to the lower stratosphere. Therefore, a stable vortex configuration also takes a long time as the composition must be stable before the temperature structure can attain its nominal winter state.

We predict that Titan's south polar vortex will continue to evolve in its current configuration until it mirrors the northern winter pole state at the start of the Cassini mission (2004–2008)[2,3,11], where stratopause temperatures of 200 K were observed in northern mid-winter. This is supported by the most recent 2016 and 2017 observations, which show the mesospheric hot-spot is returning. Cassini's incredible voyage through the Saturnian system ended on 15 September 2017 when it entered Saturn's atmosphere, making these the last measurements of Titan's winter pole until another mission can return to the Saturn system.

## Methods

**Observations.** All spectra analysed in this paper were observed with CIRS instrument[25]. CIRS is a Fourier transform spectrometer with an adjustable spectral resolution from 0.5 to 15 cm$^{-1}$. The full spectral range of CIRS covers 10–1500 cm$^{-1}$, which is split between three focal planes (FP1, FP3, FP4). Here we use FP4 (1100–1500 cm$^{-1}$) to study temperature from the CH$_4$ $\nu_4$ emission and FP3 (600–1100 cm$^{-1}$) to study trace gas emission. Both FP3 and FP4 consist of a linear array of 10 detector elements, each with a field of-view of 0.27 mrad.

Our primary observation mode is limb viewing, where CIRS observes Titan's horizon with a spectral resolution of 0.5 or 14 cm$^{-1}$. These types of observations have high vertical resolution of ~40 km (~1 atmospheric scale height) and are ideal for atmospheric profiling of temperature and composition. Low spectral resolution 14 cm$^{-1}$ CIRS spectra can be taken rapidly, allowing many latitudes to be observed during a single flyby, which is ideal for determining altitude-latitude temperature cross-sections. High spectral resolution 0.5 cm$^{-1}$ CIRS spectra are more time consuming to measure as CIRS' scan mirror needs to travel further, so high spectral resolution data are obtained in a sit-and-stare or integration mode and only observe a single latitude. These data have high signal-to-noise and can resolve distinct trace gas emission peaks, so are used to determine temperature and composition. Details of limb viewing observations are summarised in Supplementary Data 1. A sequence of six limb integrations close to 80°S provide an accurate probe of the south polar vortex temperature and composition evolution. Both types of limb observation require ~4 h of stable viewing of the limb, so must be taken ~8 h before or after closest approach. This means that the south polar region can only be viewed in this way when Cassini's orbit lies close to Titan's orbital plane. High inclination orbits only allow limb viewing of the lower latitudes.

Our secondary observation mode is nadir (downward) viewing at moderate 2.5 cm$^{-1}$ spectral resolution. Nadir observations allow mapping of the polar region in order to assess the horizontal extent of the vortex. These observations require moderate spatial resolution so can be observed when Cassini is further from Titan, around 10–20 h either side of closest approach. A full hemisphere map requires ~4 h to build up, so a stable view of the polar region is required. Therefore, nadir polar mapping is only possible when Cassini has an inclined orbit. Details of nadir viewing observations are summarised in Supplementary Data 2.

**Data analysis.** The data analysis methods are detailed in our previous Cassini CIRS limb[3,21] and nadir[8,50] studies, but are briefly summarised below.

To increase the signal-to-noise of the observations we combined multiple spectra prior to analysis. For the limb data we fitted a smooth curve to the radiance-altitude profile at each wavenumber using a cubic b-spline with a knot spacing equal to the field of view size[3,21,51]. For nadir data, we averaged spectra into 10° latitude bins and only included data that subtended an angle at Titan's centre of ≤20° to the plane containing the sub-spacecraft point and Titan's rotation axis. This ensured that the range of emission angles within each bin was ≤20° and allowed the observations to be modelled effectively with a single mean emission angle[8]. Prior to averaging the latitudes were re-projected to the level of peak emission (~150 km) to limit latitude bias due to viewing geometry[8]. For both limb and nadir data, anomalous spectra were rejected using a z-test, with a criteria that spectra more than seven standard deviations from the mean were considered outliers.

Temperature and composition were inverted from the spectra using the NEMESIS retrieval tool[31]. NEMESIS uses an iterative non-linear inversion scheme to minimise the misfit between the synthetic and observed spectra, while simultaneously minimising the deviation from an a-priori atmospheric state based on previous observations. The a-priori temperature profile was based on the Huygens HASI measurements for pressures greater than 56 mbar[36], CIRS limb soundings for pressures less than 2 mbar[2], and a linear interpolation (in log pressure) in between. Trace gas a-priori profiles were assumed to be uniform above the condensation level and relative abundances were based on Huygens GCMS measurements[52,53] or previous CIRS results[1,5,54,55]. The main photochemical aerosol was assumed to have the spectral properties defined in[56] and a 65 km scale height[57]. Spectroscopic parameters of the atmospheric gases were primarily based on the HITRAN[58] and GEISA[59] databases. CIA absorption coefficients were taken from the standard tabulations in[60–65].

For both nadir and limb observations we used a two-stage retrieval process following our previous studies[3,8,21,50]. In the first stage, a continuous atmospheric temperature profile was retrieved using the FP4 $\nu_4$ CH$_4$ emission band. The abundance of CH$_4$ is known from the Huygens GCMS measurements to be 1.48% in the stratosphere and mesosphere[53]. A smoothing equivalent to one atmospheric scale height was applied during the inversion, which is comparable to the nadir contribution function half-width and the limb field of view size. For the limb data an altitude shift on the tangent heights were also inverted for to allow for Cassini pointing errors of ≤0.1 mrad. In the second stage the abundances of the trace gases and the aerosol were fitted. For the nadir data, only a scale factor on the a-priori uniform profiles could be obtained as nadir data has very limited vertical composition information. For the 0.5 cm$^{-1}$ resolution limb data, a continuous profile of each gas was retrieved, again using a smoothing equivalent to one atmospheric scale height. At the same time the main photochemical aerosol was scaled to fit the observed limb and nadir spectra during the composition retrievals. The CIRS limb data provide information on temperature and composition in the 2 mbar–1 μbar pressure range (150–500 km).

**Radiative balance model.** The radiative balance model model[42] splits the atmosphere into 49 layers, spaced equally in log-pressure from the surface to 0.2 μbar (~600 km). The opacity of each layer is calculated using a radiative transfer calculation that includes contributions from trace gases, aerosol, and collision-induced absorption (CIA) of N$_2$–CH$_4$–H$_2$ pairs. Temperature, aerosol, C$_2$H$_2$, C$_3$H$_4$, C$_4$H$_2$, C$_6$H$_6$, HCN, and HC$_3$N profiles were taken from this study, C$_2$H$_6$ was from[22], CO and CO$_2$ were from[54], C$_3$H$_8$ and C$_2$H$_4$ were nominal south-polar values from[55], and CH$_4$ and H$_2$ were from[53].

In addition to the main photochemical aerosol[56], we also included contributions from the nitrile aerosol peaking at 160 cm$^{-1}$ and centred at 90 km[66]. Additionally, in 2015 and 2016 we included the opacity of the 220 cm$^{-1}$ aerosol feature, labelled Haze B by[67] and centred at 140 km. These extra aerosol components affect the cooling rates significantly in the lower stratosphere but have a negligible effect above 180 km.

Finally we consider the south polar clouds observed by VIMS[23] and ISS[24]. The 300 km altitude HCN ice cloud seen by VIMS is optically thin at 0.9 μm with $\tau = 0.01$–$0.07$[23] whereas the entire south polar cloud system has an estimated total optical thickness of 0.6–2.6 at 0.89 μm from ISS[24]. There is also evidence that both cloud altitude and opacity are variable with time. These clouds will reduce stratospheric insolation following their appearance in early 2012. However, we do not have high spectral resolution limb data during 2012–2014, and by 2015 when we do have limb data again, the clouds are in darkness and stratospheric insolation is negligible anyway. Therefore, for simplicity, we do not include the effect of the VIMS[23] and ISS[24] clouds in our model. As a result it is possible that we underestimate the long-wave cooling rate for altitudes of 300 km and below as our modelled aerosol features at 160 and 220 cm$^{-1}$ may not be fully representative of the south polar region.

The long-wave radiative cooling in each layer is given by $-(g/C_p)dF/dp$, where $F$ is the infrared thermal flux integrated from 0 to 1500 cm$^{-1}$ using the radiative transfer equation.

**Code availability.** The radiative transfer code and radiative balance model are fully described in ref. [31,42]. Plots were generated using the Generic Mapping Tools software[68].

**Data availability.** The Cassini CIRS data are available from NASA's Planetary Data System (https://pds.nasa.gov).

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

## Acknowledgements

This work was funded by the UK Science and Technology Facilities Council, the UK Space Agency, and the NASA Cassini Mission.

## Author contributions

N.A.T. conceived the study, performed the radiative transfer analysis, and wrote the initial manuscript. B.B. developed the radiative balance model. S.V. performed independent radiative transfer analyses to confirm the results. P.G.J.I./N.A.T. developed and maintained the radiative transfer code used for the main analysis. All authors contributed to the interpretation, in addition to editing and improving the final manuscript.

## Additional information

**Competing interests:** The authors declare no competing financial interests.

