## [Peer Review File · Nature Communications]

Reviewer #1 (Remarks to the Author):

As I already wrote in my first review, this paper presents a comprehensive of the formation and evolution of the south polar vortex in Titan's middle atmosphere. The major claims of this paper are that Titan's polar vortex in the middle atmosphere counter-intuitively experiences cooling after vernal equinox and this is likely to be a consequence of strong radiative cooling associated with an accumulation of trace species at the spring pole. With the additional information and explanation given by the authors in the rebuttal as well as in the manuscript, I see that the major part of this work is indeed novel and original.

In the revised manuscript the authors addressed my major and minor comments in a satisfactory fashion. Most importantly, the revised temperature plots are now consistent with the text. A possible dynamic origin of the observed temperature evolution is discussed. The discussion shows that waves cannot explain the observed temperature anomaly. I can accept this discussion. Furthermore, I am glad to see that the revised paper shows all relevant figures in the main part. This makes it easier to follow the content and recognize the significance of this work.

I believe this paper is worth publishing in Nature Communications and I can recommend publication as is.

Reviewer #2 (Remarks to the Author):

Reviewer 2 raises the possibility that there might be a more purely wave dynamical cause of the observed sudden cooling, perhaps an analog of Earth's sudden stratospheric warmings (SSW) that are a common feature of our Northern Hemisphere. I do feel that this is an interesting possibility. The authors interpreted this suggestion to mean lateral mixing by waves (due to tides or barotropic instability, based on the references they cite). But that is not what sudden stratospheric warmings are about. They are about planetary scale Rossby waves propagating *vertically* up from the troposphere to the stratosphere and dissipating there. This causes the magnitude and/or direction of the polar jet to change, and once this happens, it changes the ability of the waves to propagate upward, and adjustments take place to bring the temperature field into balance with the new wind field, causing the warming. (Whether the anomaly is warming or cooling would depend for another planet on the details of the original mean wind field and of the waves.) So it is all about the atmosphere's changing ability to serve as a waveguide for upward propagating waves, not about horizontal wave transports.

It hard to know whether any such phenomenon is occurring on Titan. Indeed, SSWs aren't completely understood on Earth and are not easy for models to reproduce. And I'd further suggest that such a study for Titan would be worthy of a separate paper since it would be a non-trivial question to answer. So I'd give the authors a pass on this, but I'd prefer to see them acknowledge the possibility with an appropriate sentence

Re: Revision of NCOMMS-17-18098A
“The formation and evolution of Titan’s winter polar vortex”.
N. A. Teanby et al.

Response to Reviews

Reviewer 1

Is now happy.

Reviewer 2

Had a final point about sudden stratospheric warmings. We have added the following into the main text.

“However, there is also likely to be an unknown contribution from tropospherically generated gravity wave breaking or interaction of tropospherically forced planetary waves with the vortex. For example, on Earth sudden stratospheric polar warming can be generated by vertically propagating Rossby waves, which dissipate in the stratosphere and can change the strength and direction of the vortex winds, leading to sudden adjustments in the temperature field [1, 2]. It is unclear if similar mechanisms are important on Titan and we do not consider them further here. ”

References

- [1] Matsuno, T. A Dynamical Model of the Stratospheric Sudden Warming. *J. Atmos. Sci.* **28**, 1479–1494 (1971).
- [2] Andrews, D. G., Holton, J. R. & Leovy, C. B. *Middle Atmosphere Dynamics* (Academic Press, Orlando, 1987).